# Antibacterial and Anti-Inflammatory Effects of Apolipoprotein E

**DOI:** 10.3390/biomedicines10061430

**Published:** 2022-06-17

**Authors:** Manoj Puthia, Jan K. Marzinek, Ganna Petruk, Gizem Ertürk Bergdahl, Peter J. Bond, Jitka Petrlova

**Affiliations:** 1Division of Dermatology and Venereology, Institution of Clinical Sciences, Lund University, SE-22184 Lund, Sweden; manoj.puthia@med.lu.se (M.P.); ganna.petruk@med.lu.se (G.P.); 2Bioinformatics Institute (A*STAR), Singapore 138671, Singapore; marzinekj@bii.a-star.edu.sg (J.K.M.); peterjb@bii.a-star.edu.sg (P.J.B.); 3Division of Infection Medicine, Institution of Clinical Sciences, Lund University, SE-22184 Lund, Sweden; gizemertrk@yahoo.com; 4Department of Biological Sciences, National University of Singapore, 14 Science Drive 4, Singapore 117543, Singapore

**Keywords:** apolipoprotein E, antimicrobial peptides, Gram-negative bacteria, host defense, innate immunity, aggregation

## Abstract

Apolipoprotein E (APOE) is a lipid-transport protein that functions as a key mediator of lipid transport and cholesterol metabolism. Recent studies have shown that peptides derived from human APOE display anti-inflammatory and antimicrobial effects. Here, we applied in vitro assays and fluorescent microscopy to investigate the anti-bacterial effects of full-length APOE. The interaction of APOE with endotoxins from *Escherichia coli* was explored using surface plasmon resonance, binding assays, transmission electron microscopy and all-atom molecular dynamics (MD) simulations. We also studied the immunomodulatory activity of APOE using in vitro cell assays and an in vivo mouse model in combination with advanced imaging techniques. We observed that APOE exhibits anti-bacterial activity against several Gram-negative bacterial strains of *Pseudomonas aeruginosa* and *Escherichia coli*. In addition, we showed that APOE exhibits a significant binding affinity for lipopolysaccharide (LPS) and lipid A as well as heparin. MD simulations identified the low-density lipoprotein receptor (LDLR) binding region in helix 4 of APOE as a primary binding site for these molecules via electrostatic interactions. Together, our data suggest that APOE may have an important role in controlling inflammation during Gram-negative bacterial infection.

## 1. Introduction

Apolipoprotein E (APOE) is a 299-residue glycoprotein that functions as a key regulator of cholesterol and lipid levels in the blood plasma and brain. APOE is an exchangeable apolipoprotein that is a protein component of high-density lipoproteins (HDL), very low-density lipoproteins (VLDL), and low density (LDL) remnant particles, which all have atherogenic effects. Multifunctional APOE has been reported to affect cholesterol efflux, coagulation, macrophage function, oxidative processes, central nervous system physiology, cell signaling, and inflammation [1].

In humans, APOE exhibits polymorphism, with the three alleles of APOE genes located at chromosome 19: APOE2, APOE3 and APOE4. These three isoforms only differ in amino acids composition at positions 112 and 158. The most frequent isoform APOE3 (78% of human population) has cysteine at residue 112 and arginine at residue 158; in contrast, APOE2 (8%) has cysteine at both sites, and APOE4 (14%) has arginine at both sites [2,3].

Multiple lysine and arginine amino acids in the 130–150 residues region of APOE are essential for the binding of low-density lipoprotein receptors (LDLR), as well as heparin [4,5]. The N-terminal domain of APOE forms an amphipathic four-helix bundle with LDLR- and heparin-binding sites within helix 4. The four helices are organized in an antiparallel fashion. The C-terminal domain of APOE is arranged as amphipathic α-helices, which are responsible for host lipid/lipoprotein-binding capability and protein–protein interactions [3].

It has been shown that APOE = driven peptides from the receptor- and heparin-binding region of APOE exhibit antibacterial, antiviral and immunomodulatory effects [6,7,8,9]. Moreover, we have previously reported that full-length APOE has antibacterial activity against Gram-negative bacteria, both in vitro and in vivo [10].

Although the role of lipid-poor/free APOE in the skin is not fully known, we hypothesized that the APOE may have a neutralizing effect on Gram-negative bacterial infection. In this study, we demonstrated the antibacterial activity of APOE against multiple Gram-negative bacterial strains in vitro. We also detected specific interaction and the formation of APOE aggregates upon lipopolysaccharide (LPS) challenge, which were also confirmed by molecular dynamics (MD) simulations. Moreover, we showed an immunomodulatory effect of APOE in cultured cells and animal models during LPS challenge.

## 2. Materials and Methods

### 2.1. Bacterial Strains

*E. coli* (25922 and 700928) and *P. aeruginosa* (27853) were purchased from American Type Culture Collection (ATCC). *P. aeruginosa* clinical strains 15159 and *P. aeruginosa* (PAO1) were kindly provided by Dr. B. Iglewski (University of Rochester).

### 2.2. Endotoxins

LPS from *E. coli* (serotype 0111:B4, cat# L3024), Lipoteichoic acid (LTA) from *S. aureus* (cat# tlrl-pslta) and LPS-EB Biotin from *E. coli* (cat# tlrl-lpsbiot) were purchased from Sigma-Aldrich. Lipid A from *E. coli* (serotype R515, cat# ALX-581-200-L002) was purchased from AH Diagnostics.

### 2.3. Proteins and Peptides

Human plasma APOE (cat# IHUAPOE) and human plasma APOA1 (cat# IRHPL0059) were purchased from Innovative Research. The thrombin-derived peptide TCP-25 (GKYGFYTHVFRLKKWIQKVIDQFGE) (97% purity, acetate salt) was synthesized by Ambiopharm (Madrid, Spain). Both APOE and Thrombin-derived C-terminal Peptide (TCP-25) contain amphipathic regions with a strong positive charge (K and L amino acid rich) and hydrophobicity (L, V, and I amino acid rich). Those two structural features are well-known to play important roles in antimicrobial, anti-inflammatory, and immunomodulatory activities of host defense molecules.

### 2.4. Cells

THP-1-XBlue-CD14 reporter monocytes (InvivoGen) were cultured in RMPI 1640-GlutaMAX-1 (Gibco, Life Technology Itd., Renfrew, UK) and the media was supplemented with 10% (*v*/*v*) heat-inactivated FBS (FBSi, Invitrogen, Waltham, MA, USA) and 1% (*v*/*v*) antibiotic-antimycotic solution (AA, Invitrogen) at 37 °C in 5% CO_2_.

### 2.5. Animals

BALB/c tg(NF-κB-RE-Luc)-Xen reporter male mice (10–12 weeks old), purchased from Taconic Biosciences, Rensselaer, NY, USA, were used for all experiments. The animals were housed under standard conditions of light and temperature and had free access to standard laboratory chow and water.

### 2.6. Viable Count Assay (VCA)

The potential antibacterial activity of APOE on *E. coli* and *P. aeruginosa* strains was explored by incubating one colony overnight in 5 mL of Todd-Hewitt (TH) medium. The next morning, the bacterial culture was refreshed and grown to a mid-logarithmic phase (OD_620nm_ 0.4). The bacteria were then centrifuged, washed, and diluted 1:1000 in 10 mM Tris buffer at pH 7.4 to obtain an approximate concentration of bacteria amounting to 2 × 10^6^ cfu/mL. Next, 50 μL of bacterial suspension was incubated with 2 μM of APOE, 2 μM of TCP-25 (used as a positive control), or buffer control (10 mM Tris buffer at pH 7.4) for 2 h at 37 °C. After 2 h, serial dilutions of the samples were plated on TH agar plates, incubated overnight at 37 °C, and followed by colony counting the next day [10,11].

### 2.7. NF-κB Activity Assay

NF-κB/AP-1 activation in THP-1-XBlue-CD14 reporter monocytes was determined after 20–24 h of incubation according to the manufacturer’s protocol (InvivoGen). Briefly, 1 × 10^6^ cells/mL in RPMI were seeded in 96-well plates (180 μL) and incubated with peptides (TCP-25 1 μM; APOE 0.5, 1 and 2 μM; APOAI 1 μM), LPS (10 ng/mL) or both overnight at 37 °C, 5% CO_2_ in a total volume of 200 μL. The following day, the activation of NF-κB/AP-1 was analyzed as the secretion of embryonic alkaline phosphatase (SEAP). The supernatant (20 μL) from the cells was transferred to 96-well plates, and 180 μL of Quanti-Blue was added. The plates were incubated for 2 h at 37 °C, and the absorbance was measured at 600 nm in a VICTOR3 Multilabel Plate Counter spectrofluorometer [12].

### 2.8. MTT Viability Assay

Sterile filtered MTT (3-(4,5-dimethylthiazolyl)-2,5-diphenyltetrazolium bromide; Sigma-Aldrich) solution (5 mg/mL in PBS) was stored in the dark at −20 °C until usage. We added 20 μL of MTT solution to the remaining overnight culture of THP-1-XBlue-CD14 reporter monocytes from the above NF-κB activity assay in 96-well plates, which were incubated at 37 °C (see above). After 2 h of incubation at 37 °C, we removed the supernatant and dissolved the blue formazan product generated in cells by the addition of 100 μL of DMSO (100%) in each well. The plates were then gently shaken for 10 min at room temperature to dissolve the precipitates. The absorbance was measured at 550 nm in a VICTOR3 Multilabel Plate Counter spectrofluorometer.

### 2.9. Mouse Model of Subcutaneous Inflammation and In Vivo Imaging

A mouse model of subcutaneous inflammation using NF-κB-RE-Luc reporter mice was used to study the immunomodulatory effects of APOE. The reporter mice carry a transgene containing six NF-κB-responsive elements and a modified firefly luciferase cDNA. The reporter gene is inducible by LPS and helps in in vivo studies of transcriptional regulation of the NF-κB gene. Mice were anesthetized using a mixture of 4% isoflurane and oxygen. Using a trimmer, hair was shaved from the back of the mouse and cleaned. Overnight culture of *P. aeruginosa* (PAO1) was refreshed and grown to mid-logarithmic phase in TH media. Bacteria were washed (5.6 × 1000 rpm, 15 min) and heat-killed for 30 min at 80 °C. Heat-killed bacteria (1 × 10^6^ cfu/mouse) and APOE (2 μM) were mixed and injected immediately without preincubation. A total of 200 μL of the mixture was injected subcutaneously either on the left or the right side of the dorsum. In vivo inflammation was then longitudinally measured by imaging bioluminescence with IVIS Spectrum (PerkinElmer Life Sciences, Boston, MA, USA). Fifteen minutes before the in vivo imaging, mice were intraperitoneally given 100 µL of D-luciferin (150 mg/kg body weight). The IVIS imaging was performed 3 and 6 h after the subcutaneous injection. The data were acquired and analyzed using Living Image 4.0 Software (PerkinElmer). Five or six mice per treatment group were used. The animal model was previously described in [12].

### 2.10. Fluorescence Microscopy

#### Live/Dead Bacteria

*E. coli* and *P. aeruginosa* viability in the aggregates was assessed by using LIVE/DEAD^®^ BacLight^TM^ Bacterial Viability Kit (Invitrogen, Molecular Probes, Carlsbad, CA, USA). Bacterial suspensions were prepared as described above for VCA. Bacterial strains were treated with 2 μM APOE, 2 μM TCP-25, or 10 mM Tris at pH 7.4. After a 1 h incubation time at 37 °C, samples were mixed 1:1 with the dye mixture, followed by incubation for 15 min in the dark at room temperature. The dye mixture was prepared according to the manufacturer’s protocol, i.e., 1.5 µL of component A (SYTO-9 green-fluorescent nucleic acid stain) and 1.5 µL of B (red-fluorescent nucleic acid stain propidium iodide) were dissolved in 1 mL of 10 mM Tris at pH 7.4. Five μL of stained bacterial suspension were trapped between a slide and an 18 mm square coverslip. Ten view fields (1 × 1 mm) were examined from three independent sample preparations using a Zeiss AxioScope A.1 fluorescence microscope (objectives: Zeiss EC Plan-Neofluar 40×; camera: Zeiss AxioCam MRm; acquisition software: Zeiss Zen 2.6 [blue edition]) [10].

### 2.11. Thioflavin T Assays (ThT)

Amyloid formation was determined using the dye Thioflavin T (ThT). Thioflavin T preferentially binds to the β-sheet structures of amyloidogenic proteins/peptides. For examination of the concentration dependence of the aggregation, we incubated APOE (2 μM) and LPS from *E. coli* (100 μg/mL) in buffer (10 mM Tris, pH 7.4) for 30 min at 37 °C before measurements. Two hundred microliters of the materials were incubated with 100 μM ThT for 15 min in the dark (ThT stock was 1 mM stored in the dark at 4 °C). We measured ThT fluorescence using a VICTOR3 Multilabel Plate Counter spectrofluorometer (PerkinElmer, Boston, MA, USA) at an excitation of 450 nm, with excitation and emission slit widths of 10 nm. The baseline (10 mM Tris pH 7.4 and LPS) was subtracted from the signal of each sample [12].

### 2.12. Transmission Electron Microscopy (TEM)

APOE was visualized using TEM (Jeol Jem 1230; Jeol, Tokyo, Japan) in combination with negative staining after incubation with LPS and lipid A or buffer. Images of endotoxins (100 μg/mL) in the presence or absence of APOE (2 μM), were taken after incubation for 30 min at 37 °C. For the mounted samples, ten view fields were examined on the grid (pitch 62 μm) from three independent sample preparations. Samples were adsorbed onto carbon-coated grids (Copper mesh, 400) for 60 s and stained with 7 µL of 2% uranyl acetate for 30 s. The grids were rendered hydrophilic via glow discharge at low air pressure. The size of aggregates was analyzed as the mean of gray value/μm ± standard deviation (SD) by ImageJ 1.52k, after all the images were converted to 8-bit and the threshold was manually adjusted [10].

### 2.13. Biacore Analysis

Binding experiments were carried out by using Biacore X100 (Cytiva Life Sciences, Uppsala, Sweden) with control software version of v.2.0. Sensor chip CM5 or SA (Cytiva Life Sciences, Uppsala, Sweden) were used as the gold surface for the immobilization and all the assays were carried out at 25 °C. An amine coupling kit (Cytiva Life Sciences, Uppsala, Sweden) which contained EDC [1-Ethyl-3-(3-dimethylamino-propyl)carbodiimide] (75 mg/mL), NHS (N-hydroxysuccinimide) (11.5 mg/mL) and ethanolamine (1 M, pH 8.5) or an biotin CAPture kit (Cytiva Life Sciences, Uppsala, Sweden) were used for the covalent immobilization of the ligands (APOE and LPS-biotin) on the gold surface.

Before starting the immobilization procedure, the chip (CM5 or SA) was docked into the instrument and the chip surface was activated following the EDC/NHS or/SA protocol with HBS-P or -EP buffer (0.01 M HEPES pH 7.4; 0.15 M NaCl; 0.005% *v*/*v* Surfactant P20) or as the running buffer. The ligand at a concentration of 0.01 mg/mL (in 10 mM acetate buffer, pH 5.0) was injected for 7 min (flow rate: 10 µL/min) followed by a 7 min (flow rate: 10 µL/min) injection of 1.0 M ethanolamine in order to deactivate excess reactive groups. The immobilization procedure was completed after the targeted immobilization level (≈1200 RU for APOE and ≈600 RU for LPS-biotin) was reached. Only the flow channel_2 (active channel) was used for the ligand immobilization, while the flow channel_1 (reference channel) was used as a reference to investigate non-specific binding. The subtracted channel (flow channel_2–flow channel_1) was used to evaluate the results of the analysis.

Analytes (LPS/LipA/LTA or apolipoproteins) were injected into the active (Fc_2) and reference channels (Fc_1) in concentration ranges between 0 mg/mL and 2 mg/mL, respectively. Triplicate injections were carried out for each concentration. The association time was set to 120 s while the dissociation time was kept for 600 s. The flow rate was set to 10 µL/min and 10 mM glycine-HCl (pH 2.5) was used as the regeneration buffer. To evaluate the analysis, the parameters were determined using Biacore Evaluation Software v.2.0 in equilibrium binding analysis, which was performed by plotting the RU values measured in the plateau for each concentration and fitting the data to the steady state affinity [13,14].

### 2.14. Blue Native-Polyacrylamide Gel Electrophoresis and Western Blot

Twenty-one µL of APOE (2 µM) was mixed with either 10 mM Tris as control, endotoxins (100 μg/mL) or heparin sodium salt (500 µg/mL, Sigma-Aldrich, St. Louis, MO, USA). Samples were incubated for 30 min at 37 °C before mixing with loading buffer (4 × Loading Buffer Native Gel, cat#BN2003, Life Technologies, Carlsbad, CA, USA), and, subsequently, 28 µL was loaded onto 4–16% Bis-Tris Native Gels (cat#BN1002BOX, Life Technologies). Samples were run in parallel with a marker (Native Marker Unstained Protein Standard, cat#LC0725, Life Technologies) at 150 V for 100 min. Gels were run in duplicates for each experiment: one for gel analysis after de-staining from Coomassie and subsequent staining with Gel Code Blue Safe Protein (cat# 1860983, Thermo Scientific, Waltham, MA, USA), while the other was transferred to a 0.2 μm Polyvinylidene fluoride (PVDF) membranes (Trans Blot Transfer Pack, cat #1704156, Bio-Rad, Hercules, CA, USA) via a Trans Turbo Blot system (Bio-Rad). Thereafter, the membrane was de-stained with 70% ethanol and blocked with 5% milk in 1× PBS-Tween (PBS-T) for 30 min at room temperature. The membrane was incubated with mouse mAb anti-human APOE (cat#ab1906, Abcam, Cambridge, UK) at a concentration of 1 µg/mL, diluted in 1% fat-free milk in 1× PBS-T overnight at 4 °C. APOE and its high-molecular weight complexes were then detected using a secondary rabbit anti-mouse polyclonal antibody that was conjugated to horseradish peroxidase conjugate (HRP) (cat#P0260, Dako, Glostrup, Denmark) (diluted 1:1000 in 1× PBS-T complemented with 5% milk) after incubation for 60 min at room temperature. PBS-T was used to wash the membrane after each step (3 × 10 min), and the last wash after the secondary antibody was performed five times. The bands were revealed by incubating the membrane in the developing substrate (Super Signal West Pico PLUS Chemiluminescent Substrate, cat#34580, Thermo Scientific). The signal was acquired by a Chemi-Doc (Bio-Rad) system. All the experiments were performed at least three times [10].

### 2.15. Slot-Blot Assay

We used a slot-blot assay to detect the interaction between APOE and LPS. APOE (1 μg per well) was bound to the nitrocellulose membrane (Hybond-C, GE Healthcare (Chicago, IL, USA), Biosciences) after pre-soaking in 10 mM Tris, pH 7.4. The membrane was incubated in the blocking solution (2% BSA in PBS-T, pH 7.4) for 1 h at room temperature and subsequently incubated with 20 µg/mL biotinylated LPS (LPS-EB Biotin, InvivoGen, San Diego, CA, USA) for 1 h at room temperature. Next, the membranes were washed three times for 10 min in PBS-T and incubated with streptavidin-HRP (Thermo Scientific, Rockford, IL, USA). Binding was detected using peroxide solution and a luminol/enhancer solution (1:1 *v*/*v*) (SuperSignal West Pico Chemiluminescent Substrate, Thermo Scientific). To test for competitive inhibition of peptide binding to LPS, we also performed binding studies in the presence of unlabeled heparin (6 mg/mL). Signal was acquired by a Chemi-Doc (Bio-Rad) system. All the experiments were performed at least four times [15].

### 2.16. Blood Stimulation Assay and ELISA

Fresh venous blood was collected in the presence of lepirudin (50 mg/mL) from healthy donors. The blood was diluted 1:4 in RPMI-1640-GlutaMAX-I (Gibco) and 1 mL of this solution was transferred to 24-well plates and stimulated with 0.05 or 0.1 ng/mL LPS in the presence or the absence of APOE (10, 50 and 100 nM), APOAI (100 nM) or TCP-25 (1000 nM). After 24 h of incubation at 37 °C in 5% CO_2_, the plate was centrifuged for 5 min at 1000× *g* and then the supernatants were collected and stored at −80 °C before analysis. The experiment was performed at least four times by using blood from different donors each time.

The cytokines TNF-α and IL-1β were measured in human plasma obtained after the blood stimulation experiment described above. The assay was performed by using human inflammation DuoSet^®^ ELISA Kit (R&D Systems, Minneapolis, MN, USA) specific for each cytokine, according to the manufacturer’s instructions. Absorbance was measured at a wavelength of 450 nm. Data shown are mean values ± SEM obtained from at least four independent experiments all performed in duplicate.

### 2.17. Molecular Dynamics Simulations

The initial structure of APOE was obtained from the protein data bank (PDB: 2L7B [16]). All systems were prepared using the CHARMM-GUI web server [17]. Either the full-length proteins (residues 1–299) or the truncated variants (residues 1–167) were modelled. Hydrogen atoms were added to the protein, while assuming neutral pH. The CHARMM36m [18] forcefield with TIP3P explicit water model [19] were used. In all systems, the protein was placed in the center of a cubic box of dimensions 9.5 × 9.5 × 9 nm^3^. Approximately 25,000 water molecules were added to each system; magnesium chloride salt was included to a concentration of ~150 mM, whilst neutralizing the overall system charge. For simulations in the presence of multiple lipid or heparin molecules, these were randomly placed around the surface of APOE, corresponding to twenty, five, or five molecules of heparin, lipid A or LPS, respectively. Blind docking of a single molecule of heparin or LPS was also performed using Vina-Carb [20], while AutoDock Vina 1.0 was used for lipid A [21], and, in each case, the top scoring pose was chosen for subsequent simulations. The heparin molecule was built as a tetrasaccharide 2-O-sulfo-±-L-iduronic acid 2-deoxy-2-sulfamido-±-D-glucopyranosyl-6-O-sulfate [22] using CHARMM-GUI Glycan Reader and Modeler [23]. Lipid A and LPS corresponded to the most frequently occurring species present in *E. coli*.

Energy minimization was performed for each system using steepest descents for ≤5000 steps with a 0.01 nm step size. Equilibration protocols were performed using simulations in the NVT followed by NPT ensembles for a total 50 ns with position restraints applied to protein backbone atoms. All unrestrained production simulations were run for 2 μs each in the NPT ensemble, using GROMACS2018 [24]. Simulations of full length APOE3 were run in triplicate. Equations of motion were integrated via the Verlet leapfrog algorithm with a 2 fs time step. All bonds connected to hydrogens were constrained with the LINCS algorithm. The cutoff distance was 1.2 nm for the short-range neighbor list and van der Waals interactions. The Particle Mesh Ewald method [25] was applied for long-range electrostatic interactions with a 1.2 nm real-space cutoff. The velocity rescaling thermostat was used to maintain the temperature at 310 K. Pressure was maintained at 1 bar using the Parrinello-Rahman barostat [26]. Simulations were performed on an in-house Linux cluster composed of 8 nodes containing 2 GPUs (Nvidia GeForce RTX 2080 Ti) and 24 CPUs (Intel^®^ Xeon^®^ Gold 5118 CPU @ 2.3 GHz) each. All snapshots were generated using VMD [27].

### 2.18. Statistical Analysis

The graphs of VCA, K_D_ constants (from SPR), ThT assay, TEM analysis, SPR analyses, BN gel analyses and slot blot analyses are presented as the mean ± SEM from at least three independent experiments. We assessed differences in these assays using one-way ANOVA with Dunnett’s multiple comparison tests or *t*-test. All data were analyzed using GraphPad Prism (GraphPad Software, Inc., La Jolla, CA, USA). Additionally, *p*-values less than 0.05 were considered to be statistically significant (* *p* < 0.05, ** *p* < 0.01, *** *p* < 0.001, and **** *p* < 0.0001).

## 3. Results

### 3.1. Antibacterial Activity of APOE In Vitro

The antimicrobial activity of APOE (2 µM) against several bacterial strains of *P. aeruginosa* or *E. coli* was analyzed using a viable count assay (VCA). We detected a significant reduction in bacterial growth, which was more than 100 folds for all *P. aeruginosa* strains (27853, 15159 and PAO1) (Figure 1A). Additionally, we observed approximately 50% of the bacterial killing of both *E. coli* strains (25922 and 700928) (Figure 1B). The bacterial killing efficiencies of APOE against *P. aeruginosa* and *E. coli* were compared to the thrombin-derived antimicrobial peptide TCP-25, which was used here as a positive control [28].

The data from VCA were confirmed by utilizing a live/dead imaging assay. The results revealed aggregation and dead bacteria upon treatment with 2 µM of APOE (Figure 2). Dead bacteria with disturbed membrane integrity were stained red and live bacteria with intact membranes were stained green.

### 3.2. Interaction of APOE with Bacterial Products

Next, we investigated the interaction of APOE with LPS and lipid A from *E. coli* by using the surface plasmon resonance system from Biacore. Lipid A is a lipid component of LPS, which anchors the endotoxin to the outer membrane of Gram-negative bacteria. The molecular interactions between APOE-immobilized sensor chip and LPS/lipid A were analyzed from binding curves (response vs. time) and K_D_ constants were calculated by the Biacore Evaluation Software v.2.0. The K_D_ constant for LPS was 53.7 ± 5.1 µM and for lipid A was 96.17± 17.8 µM (Figure 3A,B) The calibration curves corresponding to the interaction of APOE with LPS or lipid A are shown in Appendix A. Moreover, we validated the binding specificity of APOE by using LTA, which is the main constituent of the cell wall of Gram-positive bacteria *S. aureus*, as a negative control. No interaction was detected between APOE and LTA under the same conditions as in the previous experiment with LPS and lipid A (Appendix A).

To further verify the specific interaction of APOE with LPS, we immobilized LPS on the sensor chip and investigated the interaction of LPS with APOE and apolipoprotein AI (APOAI). APOAI is the major protein component of high-density lipoprotein particles, which play important roles in lipid metabolism. No binding interaction was observed between LPS and APOAI. The K_D_ constant of APOE was calculated from the binding curves (K_D_ 179.7 ± 50.1 µM). On the other hand, the K_D_ of APOAI could not be calculated due to the lack of affinity between the binding partners (Figure 3C,D).

### 3.3. Bacterial Products Induce the Formation of APOE Aggregates

To further investigate the binding capability of APOE to LPS or lipid A from *E. coli*, we used TEM. Analysis of the TEM images revealed that aggregate-like complexes of APOE were formed after exposure to both LPS and lipid A (Figure 4A,B). To verify the protein specificity of APOE, we carried out the TEM experiment with APOAI under the same conditions as above. TEM images analysis confirmed that APOAI did not form aggregate-like complexes in the presence of LPS (Appendix A). Furthermore, we observed an increase in APOE aggregation upon LPS challenge by detecting a significant increase in ThT fluorescence, which is suggestive of an increase in beta-sheet structural features typical of aggregating proteins (Figure 4C).

### 3.4. Heparin Blocks Interaction between APOE and LPS

Additionally, we confirmed that the interaction between APOE and LPS can be blocked by including heparin in the mixture, which bonded to the similar receptor-binding region of APOE as LPS. The blockage of the interaction was verified by blue Native gel, which was followed by a Western blot assay. We performed an image analysis of the most pronounced protein band on the WB membrane, which corresponded to the dimeric form of APOE (above 66 kDa). We observed that dimeric APOE is released from the APOE-LPS complex in the presence of heparin (Figure 5A). Moreover, we used a slot-blot assay to confirm that heparin was able to block APOE and LPS interaction. We observed a significant decrease in LPS-band intensity on the membrane additionally treated with heparin (Figure 5B).

### 3.5. Molecular Dynamics Simulations of APOE and Its Interactions with Heparin, Lipid A and LPS

In order to gain molecular insights into the dynamics of APOE, as well as its interactions with heparin, lipid A and LPS binding, we performed a series of explicitly solvated all-atom MD simulations. We first carried out 2 μs simulations of isolated full-length wild type APOE, performed in triplicate to enhance conformational sampling. The C-terminal domain, which partially shields the N-terminal domain LDLR binding site in crystallographic structures, was observed to be highly dynamic, with its backbone root-mean square deviation (RMSD) fluctuating between ~1 and 2 nm (Appendix A). This reflected the dissociation of the C-terminal domain from the N-terminal domain across all replicas, consistently leading to increased exposure of the highly positively charged LDLR receptor-binding region (Appendix A), located on helix 4 containing multiple lysine and arginine amino acids (residues ~130–150). This was quantified by a measurement of the solvent accessible surface area (SASA) of this region, which rapidly increased from an initial value of ~2.5 nm^2^ to a relatively constant value of ~10 nm^2^ over the remainder of the simulation (Appendix A).

In light of the above observations, and to improve sampling of ligand interactions with the N-terminal domain, all subsequent simulations with heparin, lipid A or LPS molecules were run with truncated APOE (residues 1–167). A series of 2 μs simulations were next performed in which different ligands were placed randomly in bulk water around APOE, in order to observe spontaneous assembly with functional regions of the protein surface. For comparison, blind docking of a single molecule of each ligand was also performed, and in each case the most frequently occurring pose involved the receptor binding domain; this pose was subsequently also used to initiate 2 μs simulations.

In the case of heparin, NMR studies showed that porcine heparin from intestinal mucosa is mostly composed of the trisulfated disaccharide: 2-O-sulfo-±-L-iduronic acid 2-deoxy-2-sulfamido-±-D-glucopyranosyl-6-O-sulfate [22], so a tetrasaccharide composed of two blocks of this were modeled. Over the course of the simulation, several heparin molecules spontaneously contacted the protein surface, with three of these tightly bound to the receptor binding region (Figure 6A). APOE residues which were involved in stable interactions with heparin for over 80% of the simulation time are shown in Figure 6B. These residues correspond to multiple basic amino acids located on: i) helix 4, with R134-R150 located in the receptor binding region along with K157-R158; ii) helix 3, including R90, R92, K95, R103 and R114; and iii) R167 at the C-terminus. In the alternative simulation initiated from a single docked heparin molecule, electrostatic interactions of heparin sulfate groups with the basic residues of helix 4 and R114 of helix 3 were observed to dominate (Appendix A).

In the case of lipid A (Figure 6C, Appendix A) and LPS (Figure 6E, Appendix A), binding simulations, electrostatic interactions between lipid phosphates and basic residues of APOE were dominant, further supported by hydrophobic interactions with lipid acyl tails wrapping around the protein surface. APOE residues involved in stable interactions with lipid A for over 80% of the simulation time are shown in Figure 6D. Similarly to heparin, these included multiple basic amino acids located on helix 4 (residues 134–150 of the receptor binding region and Q156-V161) and helix 3 (K95-R114), in addition to helix 2 (M64-K75). The interaction region including multiple basic residues was even more extensive in the case of the larger full LPS molecule (Figure 6F), including helix 4 (residues R136-R147), helix 3 (G105-A124), helix 2 (R15, T57-K72) as well as helix 1 (Q24-R38). Simulations of a single lipid A (Appendix A) or LPS (Appendix A) docked to the APOE receptor binding region further supported the role of basic residues from this region, binding to ligand phosphates, along with some nonpolar amino acid sidechains stabilizing lipid A tails. It should be noted that the reported simulations aimed to provide a structural description of the interaction between APOE and bacterial products such as LPS molecules in isolation to complement the corresponding surface plasmon resonance experiments and are indicative of the protein’s “scavenging” role and propensity for lipid-associated aggregation. Nevertheless, in the context of its antimicrobial activity, the observed dominance of electrostatics in our simulations is indicative of a potential binding mode to the Gram-negative outer membrane involving the initial approach of the receptor binding region to the anionic LPS moieties on the lipid bilayer surface.

### 3.6. Immunomodulatory Activity of APOE In Vitro

We next used reporter THP-1 monocytes to detect the effects on LPS-signaling by APOE. APOE (0.5–2 µM) significantly reduced the activation of NF-κB/AP-1 triggered by *E. coli* LPS (Figure 7A). The MTT viability assay did not show any significant cytotoxic effect of APOE on THP-1 cells, which suggests that the reduction in the NF-κB/AP-1 activation was due to the neutralizing effect of APOE on LPS and not by any APOE-mediated toxic effects on the cells (Figure 7A). TCP-25 peptide was used here as a positive control. Furthermore, we confirmed that APOAI did not reduce NF-κB activation and did not cause cell toxicity under similar experimental conditions as in the above experiment described for APOE (Appendix A).

Additionally, we performed a blood stimulation assay to validate the immunomodulatory effect of APOE on cytokine release by human immune cells in the blood. Here, we confirmed that APOE significantly decreased proinflammatory cytokines levels of both TNF-α and IL-1ß. Moreover, we observed that APOAI did not significantly reduce the release of both TNF-α and IL-1ß cytokines under similar conditions (Figure 7B).

### 3.7. Immunomodulatory Activity of APOE In Vivo

We next investigated whether APOE could suppress local inflammation triggered by bacteria in vivo. For this experiment, we utilized the (NF-κB-RE-Luc)-Xen reporter mouse model and studied the effects of APOE on subcutaneous inflammation induced by heat-killed bacteria *P. aeruginosa* (PAO1). We used killed bacteria to exclude the possible confounding effects of bacterial growth in vivo after subcutaneous deposition of live bacteria. Heat-killed bacteria (1 × 10^6^ cfu/mouse) were subcutaneously injected into the left or right side of mouse dorsum, either with or without 200 µL of APOE (2 µM). After injection of luciferin substrate, the luminescent signal, which corresponded to NF-κB activation, was detected by an in vivo bioimaging system (IVIS Spectrum) (Figure 8). IVIS image analysis showed a significant reduction in NF-κB activation after 3 and 6 h at the site challenged by both APOE and heat-killed bacteria, which was compared to heat killed bacteria-treatment alone. APOE alone did not yield any significant increase in NF-κB activation (Appendix A).

## 4. Discussion

In this study, we extended our previous work on the antimicrobial effect of full-length APOE. We have reported that APOE has a strong killing ability against only Gram-negative bacteria in vitro and in vivo [10]. Here, we confirmed the aggregation ability and antimicrobial activity of APOE against several Gram-negative laboratory or clinical bacterial strains of *P. aeruginosa* and *E. coli* at physiological plasma levels of protein, which are in healthy subjects between 3 and 7 mg/mL (proximately 0.9–2 μM) [29].

APOE is secreted from many cells throughout the human body, such as hepatic parenchymal cells, monocytes, macrophages, adipocytes, muscle and skin cells [30,31,32]. Although most of plasma APOE is lipid-bound, APOE also exists in lipid-free or lipid-poor forms. Lipid-free/poor apolipoproteins are generated in vivo by dissociation from the surface of lipoproteins or new protein synthesis [33]. Additionally, APOE is expressed in skin cells and possibly plays an important part in our innate defense system [30]. Although the exact role of lipid-poor/free APOE in the skin is not fully known, our previously published findings suggest that APOE may have a neutralizing effect against Gram-negative bacteria and their endotoxins [10].

It has been reported that peptides derived from the receptor-binding region of apolipoprotein E have an antimicrobial effect against Gram-positive and Gram-negative bacteria at the physiological protein level [34]. The antibacterial [9,35], anti-inflammatory [8], and LPS-binding [31] effects of APOE-derived peptides were described in the literature and summarized in our previous publication [10]. We extended our previous report by detecting the direct affinity of APOE to Gram-negative bacterial endotoxins using surface plasma resonance, which was complemented by MD to understand the interaction of APOE and endotoxin on a molecular level. Data obtained from our MD simulations support the role of multiple positively charged residues in the N-terminal domain in binding heparin and LPS, particularly located in the receptor binding region within helix 4 which are accessible on the APOE surface and not involved in intramolecular salt bridges.

Both antimicrobial and aggregating protein regions are usually rich in hydrophobic residues. Protein aggregation results in the self-assembly of misfolded proteins, which exhibit changes in the secondary structure. Aggregation of proteins has been implicated in many diseases such as amyloidosis, Alzheimer’s, and Parkinson’s disease. The mechanism behind protein aggregation is not still fully understood. Multiple lines of evidence point to inflammation, which may be curtailed in amyloidogenic processes [36]. APOE could be connected to the formation of aggregates, which is triggered by inflammatory activity, as a response to bacterial infection [37].

Toll-like receptor 4 (TLR4) agonists such as LPS and lipid A interact with apolipoprotein E by hydrophobic and electrostatic interactions, which cause protein aggregation. Our data suggest that protein aggregation is a key element in blocking LPS binding to TLR hydrophobic pockets, which leads to reduced LPS-triggered inflammatory signaling. APOE reduced NF-κB activation in human monocytes in vitro and in NF-κB-RE-luc mice in vivo, which indicates scavenging of LPS. Moreover, the specific and strong LPS-binding affinity of APOE compared to APOAI can be explained by previous reports showing that APOAI binds directly to LPS only via weak hydrophobic interactions. Additionally, the indirect LPS-binding affinity of APOAI was shown to be via LPS binding proteins [38].

Antimicrobial resistance is a serious threat to global public health that requires the acute need for the development of new therapeutic strategies [39,40,41]. The endogenous mechanism, by which pro-aggregation proteins such as APOE promote the containment of LPS-induced inflammation, is illustrated in Figure 9. We have previously described a similar mechanism of endogenous protein aggregation of thrombin C-terminal peptides trigged by bacterial endotoxins [12,15].

Here, we show that APOE (mostly APOE3 isoform) is able to scavenge LPS and dampen proinflammatory cytokines released from the human immune cell line, primary blood cells and animal models. Our observation is in good agreement with previously published in vivo studies on APOE-deficient mice, which show increased susceptibility to infection caused by Gram-negative bacteria or fungi [42]. We also showed apolipoprotein E specificity to Gram-negative bacteria and endotoxins compared to apolipoprotein AI.

## 5. Conclusions

In conclusion, we have demonstrated that lipid-poor/free APOE has antimicrobial activity against several Gram-negative bacterial strains of *E. coli* and *P. aeruginosa*. Additionally, we showed that APOE has a stronger killing ability for all strains of *P. aeruginosa*, whereas *E. coli* was less sensitive to the antimicrobial actions of APOE. The apolipoprotein interaction with endotoxins was specific to APOE protein and Gram-negative bacteria. The further investigation of the APOE interaction with endotoxins from Gram-negative bacteria led to the discovery of protein aggregation. Taken together, these results suggest that multifunctional APOE has an additional role in innate immunity during bacterial infection.

## 6. Patents

The peptide TCP-25 and variants are patent protected.

## Figures and Tables

**Figure 1 biomedicines-10-01430-f001:**
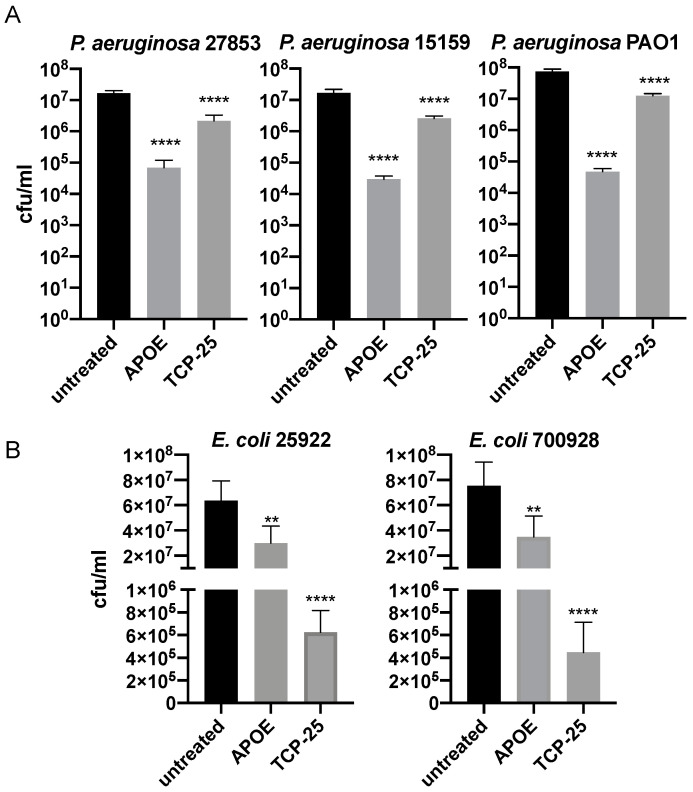
**Antimicrobial activity of APOE in vitro.***P. aeruginosa* (**A**) and *E. coli* (**B**) were incubated with 2 µM APOE for 2 h and then viable count assay was performed. Thrombin-derived peptide 2 µM TCP-25 was used as a positive control, whereas the untreated bacteria were used as negative control. Data are presented as the mean ± SEM of four independent experiments (*n* = 4). Statistical analysis was performed using a one-way ANOVA with Dunnett’s multiple comparison tests, ** = *p* ≤ 0.01 and **** = *p* ≤ 0.0001.

**Figure 2 biomedicines-10-01430-f002:**
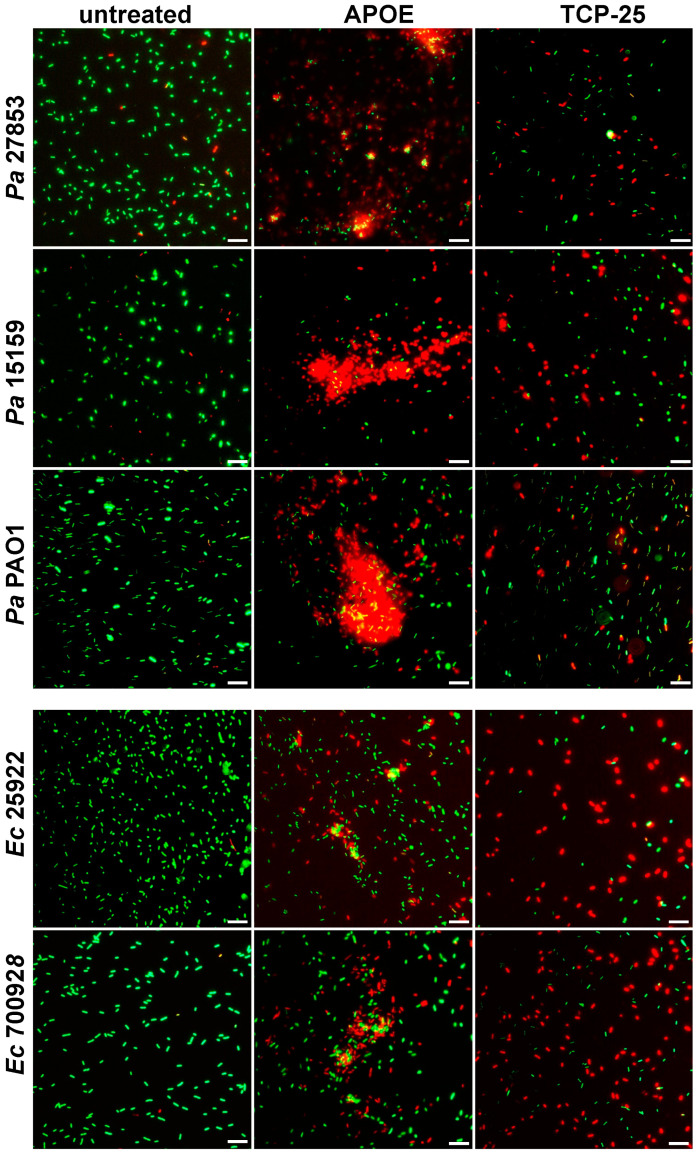
**Visualization of *E. coli* and *P. aeruginosa* viability.** Live/Dead Viability Assay of Gram-negative bacteria stimulated with 2 µM APOE, 10 mM Tris buffer at pH 7.4 (negative control) or 2 µM TCP-25 (positive control). Representative images from three independent experiments are presented (*n* = 4). At least ten individual fields were acquired per experiment. Live bacteria were stained with green SYTO 9 nucleic acid fluorescent dye, and dead bacteria were stained using red propidium iodine dye. Scale bar represents 5 µm.

**Figure 3 biomedicines-10-01430-f003:**
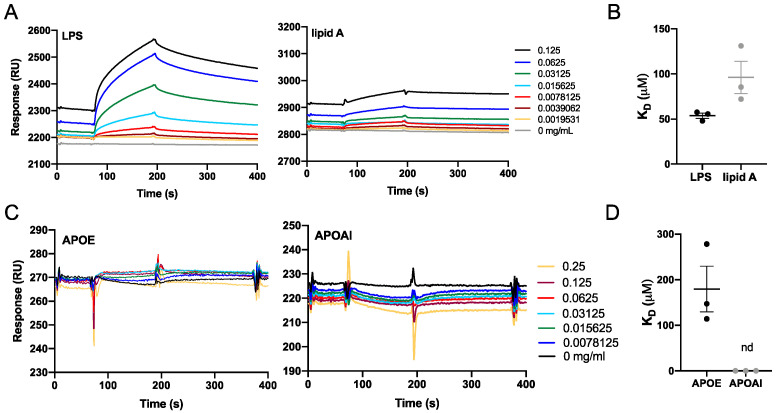
**Biacore binding analysis.** Binding interactions between the analytes and immobilized APOE protein or LPS. (**A**,**B**) Sensorgrams that show the response unit plotted as a function of time for LPS and lipid A or for APOE and APOAI. (**C**,**D**) Dissociation constants (K_D_) of protein–ligand interactions were calculated from the sensorgrams (nd = not detected).

**Figure 4 biomedicines-10-01430-f004:**
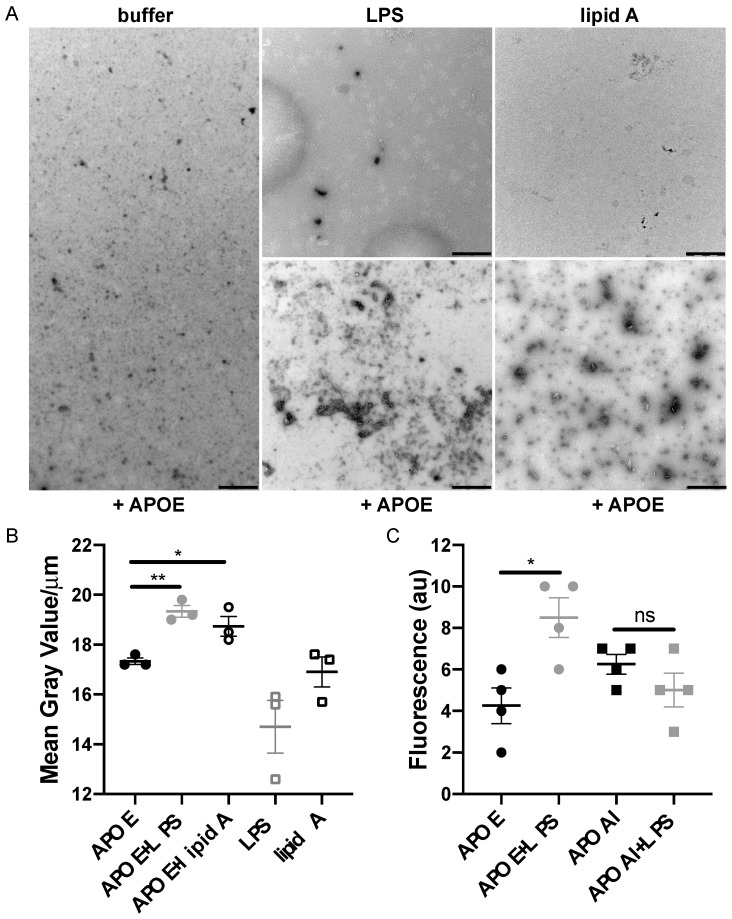
**Detection of macromolecular complexes of APOE and bacterial products.** (**A**) APOE (2 μM) was incubated with 100 μg/mL LPS or with lipid A from *E. coli* for 30 min at 37 °C. At the end of incubation, the macromolecular complexes were visualized by TEM. One representative image for each condition from three independent experiments is shown (*n* = 3). Scale bar represents 1 µm. (**B**) Analysis of the complexes of APOE and ligands following TEM. Quantification was performed using ImageJ 1.52k after all the images were converted to 8-bit, and the threshold was adjusted. The complexes of APOE and ligands are expressed as the mean of gray value/μm ± SEM. In the graph, each point represents at least ten pictures per each experiment (*n* = 3). Statistical analysis was performed using a one-way ANOVA with Dunnett’s multiple comparison tests, * = *p* ≤ 0.05 and ** = *p* ≤ 0.01. (**C**) ThT assay demonstrates aggregation of APOE in the presence of LPS from *E. coli (n* = 4). Statistical analysis was performed using a one-way ANOVA with Dunnett’s multiple comparison tests, * = *p* ≤ 0.05 and ns = not significant. Black dot = APOE, gray dot = APOE + LPS, empty circle = APOE + lipid A, empty gray square = LPS, empty black square = lipid A, black square = APOAI and gray square = APOAI + LPS.

**Figure 5 biomedicines-10-01430-f005:**
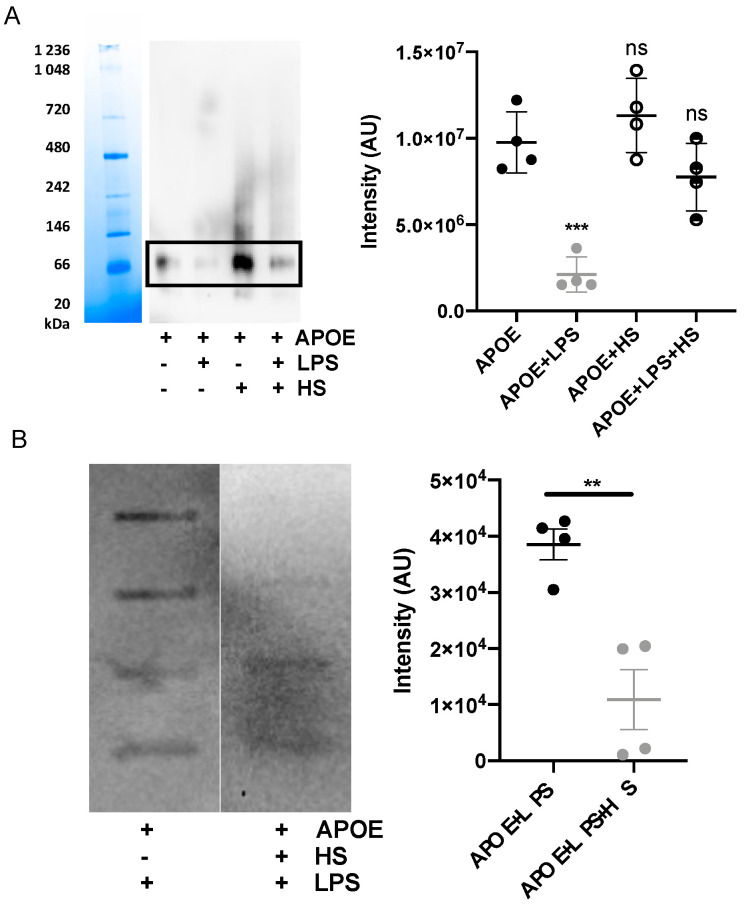
**Heparin blocks interaction of APOE with LPS.** (**A**) APOE (2 µM) was mixed with 10 mM Tris, 100 μg/mL LPS with or without heparin. Samples were incubated for 30 min at 37 °C and then run on Blue Native gel, which was followed by Western blot. One representative image from four independent experiments is shown (*n* = 4). The intensity of the bands representing dimer of APOE was measured. Statistical analysis was performed using a one-way ANOVA with Dunnett’s multiple comparison tests, *** = *p* ≤ 0.001 and ns = not significant. (**B**) A slot-blot assay demonstrated blocked binding of APOE (1 μg) to biotin-labeled LPS after the addition of heparin. Statistical analysis was performed using a *t*-test, ** = *p* ≤ 0.01.

**Figure 6 biomedicines-10-01430-f006:**
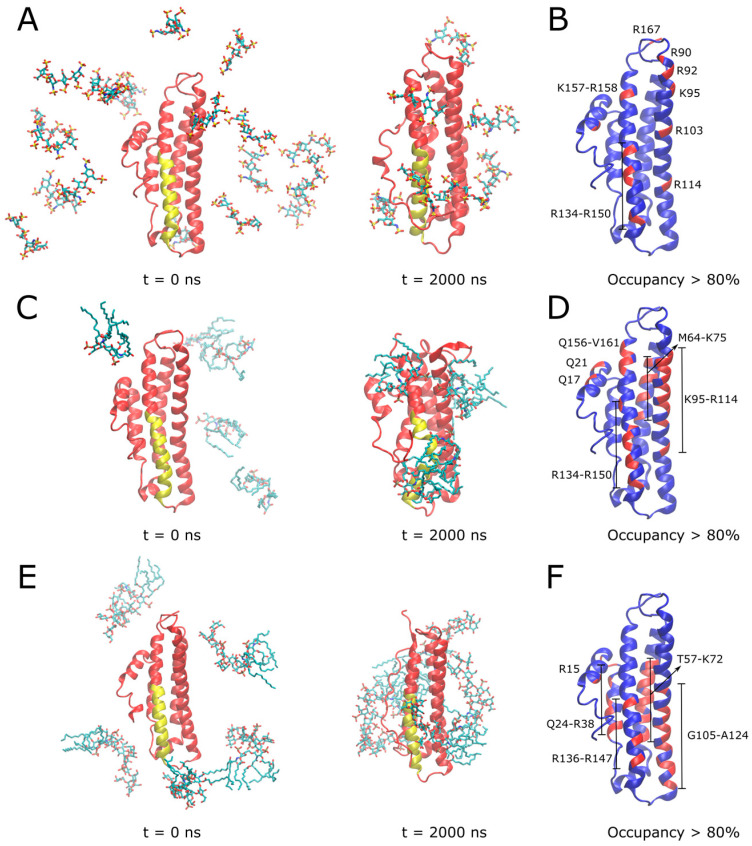
**MD simulations to investigate binding modes of heparin, lipid A and LPS with APOE.** In (**A**,**C**,**E**), initial and final simulation snapshots of systems containing truncated APOE with heparin, lipid A, and full LPS molecules are shown, respectively. In (**B**,**D**,**F**), APOE is shown in blue with a per-residue ligand occupancy (<0.3 nm cutoff distance) for >80% of total simulation sampling highlighted in red, for heparin, lipid A, and LPS systems, respectively. In (**A**,**C**,**E**), N-terminal domain is colored in red with receptor binding region on helix 4 colored in yellow. Ligand molecules are shown in CPK sticks representation (cyan—carbon; red—oxygen; blue—nitrogen; brown—phosphorus; sulfur—yellow). Simulation time points are indicated below snapshots, reported in nanoseconds (ns).

**Figure 7 biomedicines-10-01430-f007:**
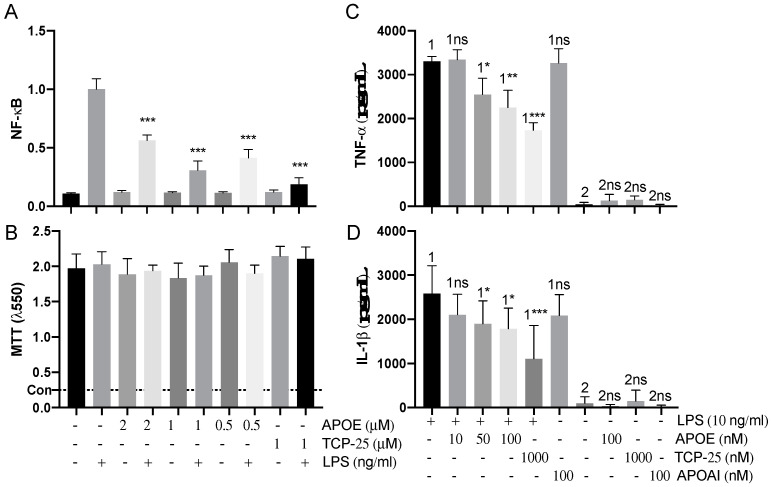
**Anti-endotoxic effect of APOE in vitro.** (**A**) THP-1-XBlue-CD14 cells were treated with APOE (0.5, 1 and 2 μM), LPS (10 ng/mL) from *E. coli*, or a combination of both. APOE yielded a significant reduction of activation of NF-κB/AP-1. *** *p* ≤ 0.001. (**B**) MTT viability assay for analysis of toxic effects of APOE on THP-1 cells. The dotted line (con) represents positive control of dead cells. The mean values of five measurements ± their SEM are shown (*n* = 5). *p* values were determined using one-way ANOVA with Dunnett’s multiple comparison test. Cytokine analysis of TNF-α (**C**) and IL-1ß (**D**) the blood collected from healthy donors at 24 h after treatment with 10 ng/mL LPS together with 10–50–100 nM APOE. Untreated blood was used as a baseline control. Blood treated with 10 ng/mL LPS together with 100 nM APOAI or 1000 nM TCP-25 were used for comparison. The mean ± SEM values of four independent experiments performed in duplicate are shown (*n* = 4). * = *p* ≤ 0.05, ** = *p* ≤ 0.01 and *** = *p* ≤ 0.001, ns = not significant, determined using two-way ANOVA with Sidak’s multiple comparisons test.

**Figure 8 biomedicines-10-01430-f008:**
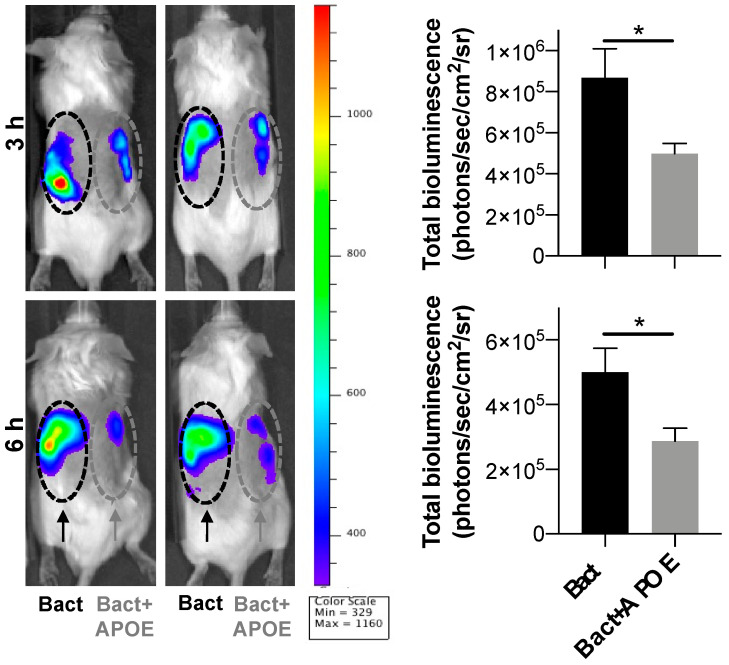
**Anti-endotoxic effect of APOE in vivo.** NF-κB activation in the NF-κB-RE-luc random transgenic mouse model was analyzed by the IVIS imaging. Heat-killed bacteria PAO1 (1 × 10^6^ cfu) (left side of the dorsum) or heat-killed bacteria (1 × 10^6^ cfu) treated with APOE (2 μM) (right side of the dorsum) were injected subcutaneously and the NF-κB response was longitudinally imaged 3 and 6 h after the subcutaneous injection. Representative images show bioluminescence at 3 and 6 h after subcutaneous injection. Bar charts show bioluminescence measured from these reporter mice. Data are presented as the mean ± SEM (*n* = 5). *p* value was determined using an unpaired *t*-test. * = *p* ≤ 0.05.

**Figure 9 biomedicines-10-01430-f009:**
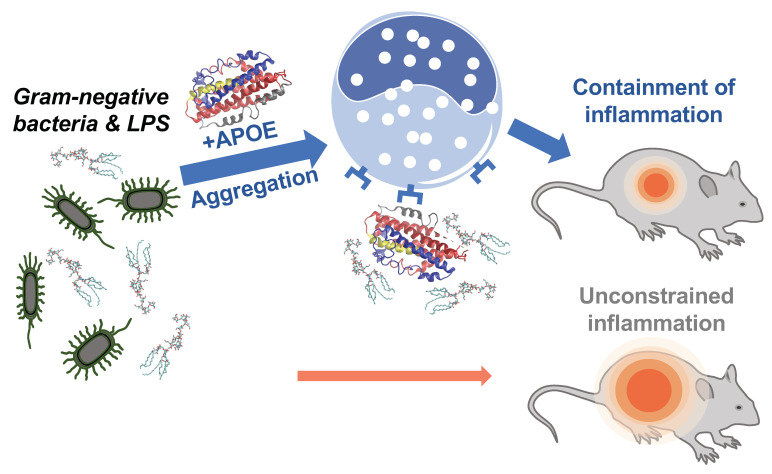
**Summary of APOE function.** Neutralizing effect of APOE on Gram-negative bacterial infection in vivo.

## Data Availability

Data is contained within the article and Appendix A.

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
