# Peer review of "Antibacterial and Anti-Inflammatory Effects of Apolipoprotein E"

_biomedicines, 2022, doi:10.3390/biomedicines10061430_

Round 1

Reviewer 1 Report

The authors have shown that lipid free apoE is able to exert antibacterial effect.  It binds to LPS and lipid A and results show that it has high affinity towards negative lipids, all characteristic of Arg-rich protein apoE or the peptides which are Arg-rich.  The authors have not properly reviewed and referenced all of the Arg-rich synthetic peptides and their reported antibacterial activities that are already described in the literature.  Two different types of apoE mimetic peptides with antibacterial and ability to bind to LPS have been reported in the literature (and the authors have failed to quote these papers) and thus, this is not novel.  The major problem with this report is, similar properties have been reported for apoA-I and its lipid complexes, which is not an Arg-rich protein (and the authors have to quote this) where the mechanism of action is different, i.e., protein-lipid-complexes act as a sink for lipid A.  Another problem in this report is given apoE-containing lipoproteins may be antibacterial (which is not tested in this manuscript) and given that lipid-free apoE may not be available in vivo, showing apoE has these properties of binding to LPS is not significant.  They have to study apoE-lipid-complexes or apoE-containing lipoproteins to attach in vivo significance for this work.  

Author Response

Dear reviewer,

Please, find enclosed our revised manuscript entitled “Antibacterial and anti-inflammatory effects of apolipoprotein E” by myself, Manoj Puthia, Jan K. Marzinek, Ganna Petruk, Gizem Ertürk Bergdahl and Peter J. Bond.

We greatly appreciate the generally positive response from the reviewers with respect to our manuscript “Antibacterial and anti-inflammatory effects of apolipoprotein E". We would like to thank the reviewers for their insightful and thorough analysis of our work and are pleased to resubmit a new version of the MS where the referee's comments have been fully addressed.

Thanks to the reviewing process, we believe our MS has been further improved. We hope you and the reviewers will now find it suitable for publication in Biomedicines. 

Responses to Reviewer's Comments:

Reviewer 

The authors have shown that lipid free apoE is able to exert antibacterial effect.  It binds to LPS and lipid A and results show that it has high affinity towards negative lipids, all characteristic of Arg-rich protein apoE or the peptides which are Arg-rich. 

Comment:

The authors have not properly reviewed and referenced all of the Arg-rich synthetic peptides and their reported antibacterial activities that are already described in the literature. 

Two different types of apoE mimetic peptides with antibacterial and ability to bind to LPS have been reported in the literature (and the authors have failed to quote these papers) and thus, this is not novel. 

Response:

We thank the reviewer for this comment. We added more information and references on the APOE mimetic peptides in the text of the manuscript. Moreover, we have previously summarized reported information on the APOE derived antimicrobial peptides in a supplementary table published in Petruk G., et al., JLR (PMID: 34019903). We clarified our previously reported finding in the text as well (Page 18, lines 887-889).

References

1.         Yao, X., Fredriksson, K., Yu, Z. X., Xu, X., Raghavachari, N., Keeran, K. J., Zywicke, G. J., Kwak, M., Amar, M. J., Remaley, A. T., and Levine, S. J. (2010) Apolipoprotein E negatively regulates house dust mite-induced asthma via a low-density lipoprotein receptor-mediated pathway. Am J Respir Crit Care Med 182, 1228-1238

2.         Croy, J. E., Brandon, T., and Komives, E. A. (2004) Two apolipoprotein E mimetic peptides, ApoE(130-149) and ApoE(141-155)2, bind to LRP1. Biochemistry 43, 7328-7335

3.         Dobson, C. B., Sales, S. D., Hoggard, P., Wozniak, M. A., and Crutcher, K. A. (2006) The receptor-binding region of human apolipoprotein E has direct anti-infective activity. J Infect Dis 193, 442-450

4.         Laskowitz, D. T., Fillit, H., Yeung, N., Toku, K., and Vitek, M. P. (2006) Apolipoprotein E-derived peptides reduce CNS inflammation: implications for therapy of neurological disease. Acta Neurol Scand Suppl 185, 15-20

5.         Laskowitz, D. T., Thekdi, A. D., Thekdi, S. D., Han, S. K., Myers, J. K., Pizzo, S. V., and Bennett, E. R. (2001) Downregulation of microglial activation by apolipoprotein E and apoE-mimetic peptides. Exp Neurol 167, 74-85

6.         Pane, K., Sgambati, V., Zanfardino, A., Smaldone, G., Cafaro, V., Angrisano, T., Pedone, E., Di Gaetano, S., Capasso, D., Haney, E. F., Izzo, V., Varcamonti, M., Notomista, E., Hancock, R. E., Di Donato, A., and Pizzo, E. (2016) A new cryptic cationic antimicrobial peptide from human apolipoprotein E with antibacterial activity and immunomodulatory effects on human cells. FEBS J 283, 2115-2131

7.         Azuma, M., Kojimab, T., Yokoyama, I., Tajiri, H., Yoshikawa, K., Saga, S., and Del Carpio, C. A. (2000) A synthetic peptide of human apoprotein E with antibacterial activity. Peptides 21, 327-330

8.         Zanfardino, A., Bosso, A., Gallo, G., Pistorio, V., Di Napoli, M., Gaglione, R., Dell'Olmo, E., Varcamonti, M., Notomista, E., Arciello, A., and Pizzo, E. (2018) Human apolipoprotein E as a reservoir of cryptic bioactive peptides: The case of ApoE 133-167. J Pept Sci 24, e3095

9.         Zhu, Y., Kodvawala, A., and Hui, D. Y. (2010) Apolipoprotein E inhibits toll-like receptor (TLR)-3- and TLR-4-mediated macrophage activation through distinct mechanisms. Biochem J 428, 47-54

Comment:

The major problem with this report is, similar properties have been reported for apoA-I and its lipid complexes, which is not an Arg-rich protein (and the authors have to quote this) where the mechanism of action is different, i.e., protein-lipid-complexes act as a sink for lipid A. 

Response:

We appreciate this comment. We added more information and reference about the affinity of apolipoprotein AI to LPS, suggesting that the specific and strong LPS-binding affinity of APOE compared to APOAI can be explained by previous reports showing that APOAI binds directly to LPS only via weak hydrophobic interactions. Additionally, the indirect LPS-binding affinity of APOAI was shown to be via LPS binding proteins (Ma J., ABBS, PMID: 15188057), (Page 19, lines 976-979).

Comment:
Another problem in this report is given apoE-containing lipoproteins may be antibacterial (which is not tested in this manuscript) and given that lipid-free apoE may not be available in vivo, showing apoE has these properties of binding to LPS is not significant. 

They have to study apoE-lipid-complexes or apoE-containing lipoproteins to attach in vivo significance for this work.  

Response:

We thank the reviewer for this comment. A major part of plasma APOE is associated with lipoproteins but APOE also exists in a lipid-free form. The precursors for the cholesterol efflux pathway in vivo are likely to be lipid-free or lipid-poor apolipoproteins, which can be generated by dissociation from the surface of lipoproteins or by de novo synthesis. Moreover, APOE is expressed by skin cells in the epidermis and apart from its role in blood, it can therefore be regarded as a part of our innate defense system. Although the exact role of lipid-free APOE in the skin is not fully known, our findings suggest that APOE has a protective effect against Gram-negative bacteria. We agree with the reviewer that we would enhance the physiological relevance of protein in plasma by studying lipid-bound APOE in the future, but it would be out of the scope of the present study, which is focused on lipid-free form of APOE (Page 18, lines 876-884).

Reviewer 2 Report

The manuscript submitted to the Biomedicines can be considered part of the work already published by the authors that is the reference 9. This manuscript can be considered a supplement of the previous work. In the article alreay published, apoE was used at 5 mM, whereas in the present manuscript, the authors have used a concentration of 2 mM, with the exception of the assay of heparin interation of apoE with LPS. Here, the authors have used 5 mM as already done in the previous work.

The conclusion of both works are very similar. In the case to be published, it is necessary to write something about the new information provided in the present work when compared to that already published by the team. I tis also necessary to explain the reasons that lead the authors to use 2 mM and not other concentrations, as well as the 5 mM apoE in the heparin interation of apoE with LPS.

Author Response

Dear reviewer,

Please, find enclosed our revised manuscript entitled “Antibacterial and anti-inflammatory effects of apolipoprotein E” by myself, Manoj Puthia, Jan K. Marzinek, Ganna Petruk, Gizem Ertürk Bergdahl and Peter J. Bond.

We greatly appreciate the generally positive response from the reviewers with respect to our manuscript “Antibacterial and anti-inflammatory effects of apolipoprotein E". We would like to thank the reviewers for their insightful and thorough analysis of our work and are pleased to resubmit a new version of the MS where the referee's comments have been fully addressed.

Thanks to the reviewing process, we believe our MS has been further improved. We hope you and the reviewers will now find it suitable for publication in Biomedicines. 

Responses to Reviewer's Comments:

Reviewer 

Comment:

The manuscript submitted to the Biomedicines can be considered part of the work already published by the authors that is the reference 9. This manuscript can be considered a supplement of the previous work. In the article alreay published, apoE was used at 5 mM, whereas in the present manuscript, the authors have used a concentration of 2 mM, with the exception of the assay of heparin interation of apoE with LPS. Here, the authors have used 5 mM as already done in the previous work.

Response:

We appreciate this comment. We clarified and added more information about the normal plasma levels of APOE in healthy subjects in the text of the manuscript, which is 3-7 mg/dl (proximately 0.9 -2 µM) (Kaneva A.M., et al., Lipids Health Dis., PMID: 23537337).

The main aim of the study was to study the antibacterial and anti-inflammatory effects of APOE under more physiologically relevant conditions (Page 18, lines 868-869). 

Comment:

The conclusion of both works are very similar. In the case to be published, it is necessary to write something about the new information provided in the present work when compared to that already published by the team. I tis also necessary to explain the reasons that lead the authors to use 2 mM and not other concentrations, as well as the 5 mM apoE in the heparin interation of apoE with LPS.

Response:

We thank the reviewer for this comment. Our previous work was focused on in vitro and in vivo anti-microbial effects of apoE (5 µM) against two Gram-negative bacterial strains. We also suggested that APOE interacts with bacterial endotoxin by detecting the structural changes of the protein. The current work showed that APOE has antibacterial activity against five clinical and laboratory Gram-negative bacterial strains at the physiological plasma level of the protein. Moreover, we observed that APOE has immunomodulatory properties in vitro and in vivo by scavenging the proinflammatory effect of LPS. We added more information about our previous work in the text: We extended our previous report by detecting the direct affinity of APOE to Gram-negative bacterial endotoxins using surface plasma resonance, which was complemented by computational modelling to understand the interaction of APOE and endotoxin on a molecular level (Page 18, lines 883-886).

All experiments in our study including heparin blockage, were done with 2 µM of APOE. We corrected the typographical error in the figure legend of figure 5 (2 µM instead of 5 µM) (Page 13, line 627). 

Reviewer 3 Report

The experiments seem to be performed appropriately and well controlled.  However several questions arise:

1. why were no Gram-positive strains used?

2. What is the rationale behind a colony-counting approach rather than a traditional broth MIC assay?

3. The authors should discuss in greater depth the correlation between the binding modes of ApoE with LipidA/LPS.  What does this indicate for a physiological environment where these molecules would be anchored into the bacterial membrane?  Specifically, how do the poses in Figure6 align with a bilayer orientation.

4. Can the authors comment on any similarities between ApoE binding and TCP25?  

More broadly, there is a question regarding the nomenclature used in the paper.  Typically, protein abbreviations are capitalized (APOE) while the gene name is lowercase (apoE).  I suggest confirming the appropriate abbreviations are being used.

https://www.ncbi.nlm.nih.gov/genome/doc/internatprot_nomenguide/#b-abbreviations-and-symbols

Author Response

Dear reviewer,

Please, find enclosed our revised manuscript entitled “Antibacterial and anti-inflammatory effects of apolipoprotein E” by myself, Manoj Puthia, Jan K. Marzinek, Ganna Petruk, Gizem Ertürk Bergdahl and Peter J. Bond.

We greatly appreciate the generally positive response from the reviewers with respect to our manuscript “Antibacterial and anti-inflammatory effects of apolipoprotein E". We would like to thank the reviewers for their insightful and thorough analysis of our work and are pleased to resubmit a new version of the MS where the referee's comments have been fully addressed.

Thanks to the reviewing process, we believe our MS has been further improved. We hope you and the reviewers will now find it suitable for publication in Biomedicines. 

Responses to Reviewer's Comments:

Reviewer 

The experiments seem to be performed appropriately and well controlled.  However several questions arise:

Comment:

  1. why were no Gram-positive strains used?

Response:

We thank the reviewer for this comment. We have previously published that APOE has no killing effect on Gram-positive bacteria in vivo or in vitro (Petruk G., et al., JLR, PMID: 34019903). For that reason, the current study is focused only on Gram-negative bacteria, which is now explained in the text of the manuscript (Page 18, lines 871-872).

Comment:

  1. What is the rationale behind a colony-counting approach rather than a traditional broth MIC assay?

Response:

We appreciate this comment. The MIC value is one of the best available parameters to know the effectiveness of antimicrobial agents against bacterial strains. MIC is quite valuable in drug development and for decision-making in clinical practice. Here, we used the Viable Count Assay (VCA) to show a proof of concept. The advantage of VCA is that it estimates the number of only viable cells that are present in a sample. VCA facilitates the identification of the number of actively growing/dividing cells in a sample thus making it more informative. We have routinely used VCA to show the antimicrobial activity of host defense peptides (Puthia M., et al., Sci Transl Med., PMID: 31894104; Petruk G., et al., JLR, PMID: 34019903). In addition, we are yet to optimize conditions for MIC for APOE. (Page 3, line 160)

Comment:

  1. The authors should discuss in greater depth the correlation between the binding modes of ApoE with LipidA/LPS.  What does this indicate for a physiological environment where these molecules would be anchored into the bacterial membrane?  Specifically, how do the poses in Figure6 align with a bilayer orientation.

Response:

We thank the reviewer for this comment. The reported simulations aimed to provide a structural description of the interaction between APOE and bacterial products such as LPS molecules in isolation, to complement the corresponding surface plasmon resonance experiments, and are indicative of the protein’s “scavenging” role and propensity for lipid-associated aggregation. Nevertheless, in the context of its antimicrobial activity, the observed dominance of electrostatics in our simulations is indicative of a potential binding mode to the Gram-negative outer membrane involving initial approach of the receptor binding region (shown in yellow in Figure 6) to the charged LPS moieties on the lipid bilayer surface. We have added text to Section 3.5 to clarify this point (Page 15, lines 724-731).

Comment:

  1. Can the authors comment on any similarities between ApoE binding and TCP25?  

Response:

We appreciate this comment. We added more information about the protein and peptide in the texts on the manuscript. Both APOE and Thrombin-derived C-terminal Peptide (TCP-25) contain amphipathic regions with a strong positive charge (K and L amino acid rich) and hydrophobicity (L, V, and I amino acid rich). Those two structural features are well-known to play important role in antimicrobial, anti-inflammatory, and immunomodulatory activities of host defence molecules. (Pages 2-3, lines 91-137)

Comment:

More broadly, there is a question regarding the nomenclature used in the paper.  Typically, protein abbreviations are capitalized (APOE) while the gene name is lowercase (apoE).  I suggest confirming the appropriate abbreviations are being used.

https://www.ncbi.nlm.nih.gov/genome/doc/internatprot_nomenguide/#b-abbreviations-and-symbols

Response:

We thank the reviewer for this comment. We have changed the nomenclature according to the reviewer’s suggestion from apoE to APOE in our manuscript.

Reviewer 4 Report

The authors extended their work regarding the anti-microbial of apoE effect against Gram-negative bacteria and showed that apoE kills Escherichia coli and Pseudomonas aeruginosa. They concluded that apoE-mediated aggregation is a scavenging mechanism that acts on Gram-negative bacteria and their products in vitro as well as in vivo.

Overall, I consider that the manuscript is valuable and it can be accepted after minor text revision (for example at Rows 277, Row 371, Row 578: delete the phrase regarding apoAI which does not make any sense at the end of the manuscript, etc).

It would be great if the authors discuss more data from the literature to extend their short Introduction Section (for example the following reviews PMID: 25157031, 28660014).

Author Response

Dear reviewer,

Please, find enclosed our revised manuscript entitled “Antibacterial and anti-inflammatory effects of apolipoprotein E” by myself, Manoj Puthia, Jan K. Marzinek, Ganna Petruk, Gizem Ertürk Bergdahl and Peter J. Bond.

We greatly appreciate the generally positive response from the reviewers with respect to our manuscript “Antibacterial and anti-inflammatory effects of apolipoprotein E". We would like to thank the reviewers for their insightful and thorough analysis of our work and are pleased to resubmit a new version of the MS where the referee's comments have been fully addressed.

Thanks to the reviewing process, we believe our MS has been further improved. We hope you and the reviewers will now find it suitable for publication in Biomedicines. 

Responses to Reviewer's Comments:

Reviewer 

The authors extended their work regarding the anti-microbial of apoE effect against Gram-negative bacteria and showed that apoE kills Escherichia coli and Pseudomonas aeruginosa. They concluded that apoE-mediated aggregation is a scavenging mechanism that acts on Gram-negative bacteria and their products in vitro as well as in vivo.

Comment:

Overall, I consider that the manuscript is valuable and it can be accepted after minor text revision (for example at Rows 277, Row 371, Row 578: delete the phrase regarding apoAI which does not make any sense at the end of the manuscript, etc).

Response:

We appreciate this comment. We have made the corrections in the text of the manuscript according to the reviewer’s suggestion (Page 7, line 356; page 12, line 579; and page 19, line 969).

Comment:

It would be great if the authors discuss more data from the literature to extend their short Introduction Section (for example the following reviews PMID: 25157031, 28660014).

Response:

We thank the reviewer for this comment. We have extended the introduction of the manuscript by adding: APOE is an exchangeable apolipoprotein that is associated with high-density lipoproteins (HDL), very low-density lipoproteins (VLDL), and low density (LDL) remnant particles, which all have atherogenic effects. Multifunctional APOE has been reported to affect cholesterol efflux, coagulation, macrophage function, oxidative processes, central nervous system physiology, cell signaling, and inflammation (White, C.R,. et al., JLR, PMID: 25157031), (Pages 1-2, lines 36-51).

Round 2

Reviewer 1 Report

Previously, it was recommended that the authors do a search on apoE mimetic peptides and antimicrobacterial activity. This was not done and thus, required changes are not presented.  I am at a loss why authors changed an accepted abbreviation for apolopiproteins as Apo or apo to APO.  The original abbreviation was correct.  The gels shown for HS interfering with apoE binding to LPS are not clear. A better quality gel is needed to make any conclusion.  They can discuss how HS is interfering with apoE binding to LPS of lipidA.  They also have to discuss why apoA-I does not have the same affinity as apoE for LPS.  However, apoA-I+DMPC complexes have been shown to bind to LPS and Lipid A. The authors have not referenced this. The EM they present shows that LPS+apoE may be forming complexes but as they say, it could also be aggregation of apoE in the presence of LPS or lipid A.  There is no literature to suggest that free apoE is present in vivo.  However, as with A-I-lipid complexes, apoE-lipid complexes can also also associate LPS and lipid A, as it is true with several of the amphipathic peptides and their lipid complexes.  Lipoproteins have also been shown to be antibacterial.  These and other references, particularly the apoE mimetic peptides or cationic peptides interacting with LPS have to be provided.  
